# Zebrafish Model of Stickler Syndrome Suggests a Role for Col2a1a in the Neural Crest during Early Eye Development

**DOI:** 10.3390/jdb10040042

**Published:** 2022-10-01

**Authors:** Antionette L. Williams, Brenda L. Bohnsack

**Affiliations:** 1Division of Ophthalmology, Ann & Robert H. Lurie Children’s Hospital of Chicago, 225 E. Chicago Ave., Chicago, IL 60611, USA; 2Department of Ophthalmology, Northwestern University Feinberg School of Medicine, 645 N. Michigan Ave., Chicago, IL 60611, USA

**Keywords:** Stickler syndrome, neural crest, eye development, anterior segment, collagen type 2 alpha 1, congenital diseases

## Abstract

Most cases of Stickler syndrome are due to autosomal-dominant COL2A1 gene mutations leading to abnormal type II collagen. Ocular findings include axial eye lengthening with vitreal degeneration and early-onset glaucoma, which can result in vision loss. Although COL2A1 is a major player in cartilage and bone formation, its specific role in eye development remains elusive. We investigated the role of Col2a1a in neural crest migration and differentiation during early zebrafish eye development. In situ hybridization, immunofluorescence, live imaging, exogenous treatments [10 μM diethylaminobenzaldehyde (DEAB), 100 nM all-trans retinoic acid (RA) and 1–3% ethanol (ETOH)] and morpholino oligonucleotide (MO) injections were used to analyze wildtype Casper (*roy−/−;nacre−/−*), TgBAC(*col2a1a*::EGFP), Tg(*sox10*::EGFP) and Tg(*foxd3::EGFP*) embryos. Col2a1a colocalized with Foxd3- and Sox10-positive cells in the anterior segment and neural crest-derived jaw. Col2a1a expression was regulated by RA and inhibited by 3% ETOH. Furthermore, MO knockdown of Col2a1a delayed jaw formation and disrupted the ocular anterior segment neural crest migration of Sox10-positive cells. Interestingly, human COL2A1 protein rescued the MO effects. Altogether, these results suggest that Col2a1a is a downstream target of RA in the cranial neural crest and is required for both craniofacial and eye development.

## 1. Introduction

Stickler syndrome is an arthro-ophthalmopathy that affects approximately 1 in 7500 and is due to abnormal collagen fibril formation [1,2]. Craniofacial abnormalities consist of micrognathia, cleft palate, and hearing loss, while ocular findings include axial lengthening of the eye with vitreal degeneration and early onset glaucoma [1,2,3,4,5,6]. The increased size of the eye results in high myopia and high risk for retinal breaks, tears, and detachments [2,3,4]. Most cases represent autosomal dominant Type I Stickler syndrome due to mutations in collagen type 2 alpha 1, which is encoded by the COL2A1 gene [2,7]. Additional types of Stickler syndrome are caused by mutations in other associated collagen genes, including COL11A1 (Type II), COL11A2 (Type III), COL9A1 (Type IV), COL9A2 (Type V), and COL9A3 (Type VI), with Types II and III being autosomal dominant and Types IV, V, and VI accounting for rare autosomal recessive forms [8,9,10,11,12].

Alpha 1 chains of type II collagen assemble to eventually form type II collagen fibers that are predominantly found postnatally in hyaline cartilage [13]. The predominance of these fibers in cartilage is consistent with the early onset osteoarthritis found in many COL2A1-deficient individuals with type I Stickler syndrome. During development, COL2A1 is expressed in cranial neural crest cells that ultimately form the mandible, maxilla, and the bones of the inner ear, which accounts for the additional micrognathia, cleft palate, and hearing loss that is also observed in Stickler syndrome [14,15].

Although type II collagen is expressed in the postnatal vitreous humor [16,17,18], the origin of the ocular pathologies associated with COL2A1 mutations is not well understood. Neonates with Stickler syndrome are already found to have myopia and increased globe size, and ultimately there is minimal progression of axial lengthening through childhood [2]. Further, infantile-onset glaucoma is also reported in Type I Stickler syndrome indicating a greater role for COL2A1 in eye development [2,5,6]. However, the role of COL2A1 in eye development has yet to be elucidated.

While cranial neural crest cells derive much of the craniofacial bone and connective tissue, a subpopulation also contributes to the sclera, cornea, iris, and ciliary body within the eye. In the current studies, we used zebrafish to first investigate the expression and regulation of *col2a1a* and then assess the function of Col2a1a in the cranial neural crest with a specific focus on the ocular component.

## 2. Materials and Methods

### 2.1. Animal Husbandry and Animal Strains

Zebrafish (Danio rerio) were raised in a breeding colony on a cycle of 14 h of light followed by 10 h of dark. The embryos were collected from natural spawning and maintained at 28.5 degrees Celsius. The embryos were subsequently staged according to morphology as previously described [19] using hours after fertilization (hpf). The transgenic strains TgBAC(*cola2a1a::EGFP*), Tg(*sox10*::EGFP), and Tg(*foxd3::EGFP*) were gifts from Gage Crump, Ph.D. (University of Southern California, Los Angeles, CA, USA), Thomas Schilling, Ph.D. (University of California, Los Angeles, LA, USA) and Mary Halloran, Ph.D. (University of Wisconsin, Madison, WI, USA), respectively. These strains were crossed into the Casper (*roy−/−, nacre−/−*) background to decrease autofluorescence and interference resulting from pigmentation [20,21]. Fish protocols were approved and performed in accordance with guidelines for the humane treatment of laboratory animals established by the Northwestern Institutional Animal Care and Use Committee (IACUC, Protocol# IS00015598) of Stanley Manne Children’s Research Institute (Chicago, IL, USA), an AALAC-accredited facility.

### 2.2. Imaging

Whole embryos were analyzed using an M205FA automated fluorescence stereo microscope (Leica Microsystems CMS GmbH, Germany, Wetzler, Germany). Images were obtained using brightfield DFC290 (Leica) and fluorescent ORCA-ER (Hamamatsu, Hamamatsu City, Japan) cameras. Sections were imaged using a LSM 880 confocal laser scanning microscope (Carl Zeiss AG, Oberkochen, Germany). Z-stacks ranging from the lateral edge of the cornea to 100 mM medial to the medial edge of the eye were obtained. The Z-stacks were deconvolved and maximally projected to obtain a single image. The images were processed using Adobe Photoshop 2021 version 22.0.0 graphics editing software (Adobe Systems Incorporated, San Jose, CA, USA) and ZEN 2.1 SP3 (black) version 14.0.0.0 microscopy software (Carl Zeiss AG). The images shown are representative of all experiments. For quantification of immunofluorescence, at least 4–6 embryos per group were included.

Ocular measurements were obtained using Photoshop version 22.0.0 graphics editing software (Adobe Systems Incorporated). The dorsal-ventral and anterior-posterior dimensions of the eye were obtained as previously described [22]. Briefly, the distance between the dorsal and ventral retinal pigment epithelia (RPE) at the equator of the eye encompassing the thickest portion of the lens and the distance between the anterior surface of the corneal epithelium and the posterior RPE near the optic stalk were measured. The average of three consecutive sections for each eye was used for statistical analysis.

Quantitative analysis was performed using Photoshop version 22.0.0 graphics editing software (Adobe Systems Incorporated). Statistical analysis was performed using Microsoft Excel 2022 version 16.64 software (Microsoft Corporation, Redmond, WA, USA), and *p* < 0.05 was considered statistically significant.

### 2.3. Pharmacological Treatments

Absolute ethanol (ETOH, Sigma-Aldrich, St. Louis, MO, USA) was administered in embryo media at 1% and 3%. Wildtype (WT) Casper strain embryos were dechorionated, and the treatments were administered from 24 to 48 hpf. Following treatment, the embryos were washed multiple times with embryo media and subsequently placed in fresh embryo media for the remainder of the time course as indicated. All-trans retinoic acid (RA; Sigma-Aldrich) and 4-diethylaminobenzaldehyde (DEAB, Sigma-Aldrich, pan-aldehyde dehydrogenase inhibitor of RA synthesis) were diluted in dimethyl sulfoxide (DMSO, Sigma-Aldrich) to a 1000× final concentration. The pharmacological treatments were initiated between 24 and 27 hpf (as described in the Results). Dose curves were conducted for each pharmacologic treatment (1, 10, 25, 100 nM RA; 5, 10, 20 μM DEAB), and final concentrations were chosen based on LD50 and consistency of phenotype (data not shown). Final concentrations were as follows: 0.1% DMSO (vehicle control), 100 nM RA, and 10 μM DEAB. Experiments used 50 to 100 embryos per treatment group and were replicated 4 to 6 times. Phenotypes were assessed at 24, 36, 48, 60, 72 and 96 hpf.

### 2.4. Morpholino Oligonucleotide Injections

A lissaminated translation blocking (5′-AGATTAGACATGCAGGTCCTAAGGG-3′) antisense morpholino oligonucleotide (MO) targeting the promoter region of the zebrafish Col2a1a gene and a lissaminated standard control (5′-CCTCTTACCTCAgTTACAATTTATA-3′) antisense morpholino oligonucleotide (Gene Tools, LLC, Philomath, OR, USA) were reconstituted in de-ionized water. Concentrations yielding consistent and reproducible phenotypes were determined for each MO. One-cell stage TgBAC(*cola2a1a::EGFP*), Tg(*sox10::EGFP*), and Tg(*foxd3::EGFP*) embryos were injected with 1 to 2 nL of MO at a concentration of 0.25 mM (2.1 ng/nL). Embryos were imaged at 24, 36, 48, and 60 hpf as described above.

Human COL2A1 protein (CC052; 100 μg at 1 mg/mL; Sigma-Aldrich) was purchased and subsequently diluted in RNAse-free water. Then, 1 to 2 nL (100–600 pg total protein) of human COL2A1 was injected alone with phenol red and coinjected with 1 to 2 nL of MO at a concentration of 0.25 mM (2.1 ng/nL) into one-cell-stage embryos. Embryos were imaged at 24, 36, 48, and 60 hpf as described above.

### 2.5. Wholemount In Situ Hybridization and Immunostaining

Staged WT Casper, Tg(*sox10::EGFP*) and Tg(*foxd3::EGFP*) zebrafish embryos were fixed in 4% paraformaldehyde overnight at 4 degrees Celsius. In situ hybridization was performed as previously described [23,24] using a digoxigenin-labeled RNA antisense probe targeting zebrafish *col2a1a* (NM_131292.1; Forward: 5′-CAACGCATAGCGTTTTGCTA-3′, Reverse: 5′-GACCTGGGTGTCCTCTGTGTC-3′). After stringency washes, the embryos were subjected to immunological detection using an anti-digoxigenin antibody conjugated to alkaline phosphatase (1:1000; Sigma-Aldrich, cat. no. 11-093-274-910). Alkaline phosphatase activity was evaluated using a colorimetric assay [Vector Blue Substrate Kit, Alkaline Phosphatase (SK-5300); Vector Laboratories, Inc., Newark, CA, USA], which is both chromogenic (blue) and fluorescent (Far Red/Cy5). For colorimetric reactions, the embryos were developed for equal amounts of time. Sense controls were also developed in parallel to ensure specific staining (data not shown). Wholemount immunostaining for green fluorescent protein (GFP) was performed using standard protocols. Briefly, the embryos were first successively dehydrated in methanol and incubated overnight at −20 degrees Celsius. The next day, the embryos were successively rehydrated to 100% 1X phosphate-buffered saline (PBS), followed by blocking with 10% sheep serum, in PBS containing 0.1% Triton X-100 (PBST) and 0.5% BSA. Next, the embryos were incubated with anti-GFP (1:100; ab6556; Abcam, Waltham, MA, USA) overnight at 4 degrees Celsius. The sections were washed with PBST and incubated with goat anti-rabbit IgG cross-adsorbed Alexa Fluor™ 488 secondary antibody (Invitrogen, Waltham, MA, USA) for at least 2 h at room temperature. The embryos were subsequently cryoprotected in successive sucrose solutions, embedded in Optimal Cutting Temperature (O.C.T) compound (Fisher Scientific, Waltham, MA, USA), and then sectioned rostrocaudally through the head in an orientation perpendicular to the spinal column to obtain transverse cephalic sections at 10 μm. The sections were mounted with ProLong™ Gold Antifade Mountant with 40,6-diamidine-2-phenylidole-dihydrochloride (DAPI; Thermo Fisher Scientific, Waltham, MA, USA), coverslipped and imaged as described above.

## 3. Results

### 3.1. Col2a1a Is Expressed in the Ocular Neural Crest during Early Development

The congenital eye anomalies and facial defects observed in Stickler syndrome likely reflect an underlying disruption of the cranial neural crest. Col2a1a gene expression in the craniofacial cartilage, ear, notochord, floor plate, hypochord and fins of zebrafish has previously been shown [25,26]. Therefore, we first examined the patterning of *col2a1a* expression in and around the developing eye in WT Casper strain zebrafish embryos during development at 24, 36, 48, 60, 72 and 96 hpf. Microscopy analysis of wholemount and section colorimetric in situ hybridization showed that *col2a1a* expression initiates within the hindbrain and notochord at 24 hpf (black arrow, Figure 1A) and progresses dorsoposteriorally and ventrally into the ocular (white arrows, Figure 1A–F) and craniofacial regions (blue arrows, Figure 1A–F; blue dashed arrows, Figure 1B’–F’), respectively, with apparent expression in the anterior segment [iris (Ir) (Figure 1C”), hyaloid vasculature (black arrows, Figure 1C”), sclera (Sc) and periocular mesenchyme (PM) (Figure 1C”), ocular fissure (yellow arrow, Figure 1C), and optic nerve (red arrows, Figure 1D’; ON, Figure 1C”)] at 48 and 60 hpf (Figure 1D”). By 96 hpf, *col2a1a*-expressing neural crest cells were almost completely migrated from the ocular region and were primarily detected in the developing jaw and pharyngeal arches (Figure 1F”). Thus, in addition to its well-characterized craniofacial localization, *col2a1a* is also expressed in neural crest-derived structures of the anterior segment during early eye development. Studies in numerous animal models have demonstrated that the transcription factors Sox10 and Foxd3 are regulators of neural crest cell migration that demarcate temporally and spatially distinct neural crest cell populations in the periocular mesenchyme and within the eye [27,28,29,30,31]. To further characterize *col2a1a* expression in the cranial neural crest, we examined its expression in Tg(*sox10::EGFP*) and Tg(*foxd3::EGFP*) transgenic lines. Wholemount colorimetric in situ hybridization, followed by sectioning, immunostaining and fluorescence confocal microscopy revealed the colocalization of *col2a1a* expression with Foxd3-positive neural crest cells in the ocular anterior segment [hyaloid vasculature (arrows), iris outflow tract (IOT), sclera (Sc), and periocular mesenchyme (PM) (Figure 1H)] and Sox10-positive neural crest cells in the developing jaw [Meckel’s cartilage (Mc), trabeculae (T), and quadrate (Q)] and PM of zebrafish at 48 hpf (Figure 1G). Thus, Col2a1a is expressed in both ocular Foxd3-positive and craniofacial Sox10-positive neural crest cell populations during early development in zebrafish.

### 3.2. Retinoic Acid Regulates Col2a1a Expression in the Ocular Neural Crest

As an essential morphogen and major regulator of cranial neural crest cells during embryogenesis, the tight control of retinoic acid (RA) mediates the migration and differentiation of these cells within the pharyngeal arches, frontonasal process, and periocular mesenchyme, which ultimately give rise to cranial bones and connective tissues as well as the structures in the ocular anterior segment [32,33,34,35,36,37]. Accordingly, we next examined the influence of alterations in RA levels on *col2a1a* expression in the ocular and craniofacial neural crest. Fluorescence microscopy analysis of wholemount colorimetric in situ hybridization analyses following treatment at 24 hpf with 10 mM N,N-diethylaminobenzaldehyde (DEAB), a selective pan-aldehyde dehydrogenase inhibitor that decreases endogenous RALDH2/3-mediated RA synthesis [38,39,40], showed significantly decreased *col2a1a* expression in the anterior segment of WT Casper zebrafish embryos at 60 hpf (Figure 2B,B’,D) compared with their DMSO control-treated counterparts (Figure 2A,A’,D). In contrast, exogenous treatment with 100 nM RA at 27 hpf significantly increased *col2a1a* expression in these neural crest-derived structures at 60 hpf (Figure 2C,C’,D). Further examination of *col2a1a* expression in 60 and 96 hpf embryos treated with DEAB and RA, revealed that the teratogenicity of both exogenous treatments during ocular and craniofacial development was considerable, showing alterations in eye size (solid and dashed circles, Figure 2A–C,E–G) and marked jaw deformities (Figure 2E–G). Taken together, these results suggest that *col2a1a* is a downstream target of RA regulation in the cranial and ocular neural crest.

### 3.3. Ethanol Decreased Col2a1a Expression in the Ocular Neural Crest and Developing Jaw 

In a previous study [28], we demonstrated that neural crest cells in the periocular mesenchyme and developing eye are less sensitive to ethanol challenge compared with their craniofacial neural crest cell counterparts that contribute to pharyngeal arch formation and jaw development. This difference explains the uncommon occurrence of anterior segment dysgenesis despite the persistent craniofacial abnormalities observed in fetal alcohol syndrome. In stark contrast, congenital eye anomalies are observed together with facial defects in Stickler syndrome Type I, reflecting an autosomal dominant mutation in COL2A1. To determine whether there are any observable differences in *col2a1a* expression in neural crest cells that derive craniofacial structures versus those that contribute to the ocular anterior segment, we treated zebrafish embryos with 1% and 3% ethanol from 24 to 48 hpf, a time frame consistent with neural crest migration into the anterior segment of the eye. Subsequent wholemount colorimetric in situ hybridization analysis of WT Casper embryos at 60 and 96 hpf and fluorescence microscopy analysis of transverse sections of the head at 60 hpf revealed the effects of ethanol treatment on ocular and craniofacial development. The results showed decreased *col2a1a* expression in the periocular mesenchyme and within the developing eyes of 3% ETOH-treated embryos (Figure 3C,C’,E) compared with that in untreated and 1% ETOH-treated embryos (Figure 3A,A’,B,B’,E). Further, ethanol treatment at this high concentration dramatically affected eye size (solid and dashed circles, Figure 3A–C) and significantly delayed jaw development, showing severe malformations of the pharyngeal arches (PA, Figure 3H) and basihyal (red arrows, Figure 3H) and Meckel’s cartilage (white arrows, Figure 3H) at 96 hpf compared with untreated and 1% ETOH-treated fish (Figure 3F,G). Interestingly, treatment with 3% ETOH from 24–48 hpf together with RA insult from 27–96 hpf restored *col2a1a* expression in the ocular anterior segment and partially rescued the teratogenic effects of 3% ethanol treatment on eye development (Figure 3D,D’,E) but did not restore *col2a1a* expression in the craniofacial neural crest (yellow asterisk, Figure 3D’) at 48 hpf and further potentiated the deleterious effects of ethanol treatment on the developing jaw (white asterisk, Figure 3I) at 96 hpf. Altogether, these results show that ethanol treatment inhibited *col2a1a* expression in both ocular and craniofacial neural crest populations in the developing embryo and disrupted eye and jaw development, with defects in both regions being more severe with 3% ETOH challenge. Notably, RA significantly restored ocular *col2a1a* expression and primarily rescued the eye defects but made the jaw defects more profound.

### 3.4. Col2a1a Knockdown in Zebrafish Mimics the Craniofacial and Ocular Defects Observed in Human Stickler Syndrome Patients

Because Col2a1a expression in the neural crest-derived anterior segment was detected and colocalization with neural crest markers Foxd3 and Sox10 was determined, we investigated the function of this protein in skeletal, craniofacial, and ocular development in zebrafish. To knockdown Col2a1a, a lissamine-tagged antisense MO targeting the promoter region of *col2a1a* was injected into TgBAC(*col2a1a::EGFP*) embryos at the single-cell stage. The highly significant loss of GFP reporter expression driven by the *col2a1a* promoter in the MO-injected TgBAC(*col2a1a::EGFP*) strain (bottom right panel, Figure 4A,B), compared with that in uninjected and standard control MO-injected embryos of the same strain (upper right and middle right panels, respectively, Figure 4A,B), provided proof of concept of the feasibility of Col2a1a knockdown. The phenotypic effects of Col2a1a MO knockdown were next examined in Tg(*sox10::EGFP*) embryos at 48 and 72 hpf. Compared with uninjected and control-injected embryos (left and middle panels, Figure 4C), MO-injected Sox10-positive embryos showed skeletal defects, including scoliosis (upper right panel, dashed line, Figure 4C) and significantly shortened body length (Figure 4D), as well as severely delayed jaw and pharyngeal arch development (lower right-most panel, asterisk, Figure 4C). Importantly, significantly smaller bulging eyes (lower left-right panel, Figure 4C,E), as measured along the anterior–posterior and dorsal-ventral axes, were also observed. These effects are reminiscent of the orofacial and ophthalmic manifestations typically observed in Stickler syndrome, a heritable human connective tissue disorder most caused by mutations in the genes encoding collagen type II (COL2A1). Thus, Col2a1a knockdown not only disrupts jaw and pharyngeal arch formation but also recapitulates ocular defects characteristic of Stickler syndrome, suggesting an important role for Col2a1a in ocular development.

### 3.5. Col2a1a MO Knockdown Differentially Affected Sox10 and Foxd3 Expression and Ocular Neural Crest Cell Migration during Early Eye Development

Further analysis of the zebrafish Col2a1a knockdown Stickler syndrome model was conducted to determine the role of Col2a1a in embryonic ocular development, specifically focusing on the neural crest-derived anterior segment. For these experiments, we examined the effects of Col2a1a MO knockdown in both Tg(foxd3::EGFP) and Tg(sox10::EGFP) zebrafish lines. Lateral live imaging showed that despite the observed ocular defects, Col2a1a knockdown did not affect Foxd3-positive cell migration into the anterior segment between 36 and 48 hpf (bold arrows, Figure 5D”,F”) compared with that in standard control MO-injected (Figure 5D’,F’) and uninjected (Figure 5D,F) embryos. However, Sox10-positive cell migration was markedly disrupted in response to Col2a1a MO knockdown, as these cells were retained in the anterior segment (asterisks, Figure 5E,E’,E”) and periocular mesenchyme (single arrows, Figure 5E,E’,E”) in contrast to their uninjected and control-injected counterparts, which showed little to no Sox10-positive cells in the anterior segment at 48 hpf. Marked differences between Foxd3 and Sox10 expression in response to Col2a1a knockdown were detected in transverse cephalic sections of MO-injected zebrafish embryos at 48 hpf. MO knockdown of Col2a1a decreased the expression of Sox10 in the ocular anterior segment neural crest, while the expression of Foxd3 in the Foxd3-positive ocular neural crest cell remained unaffected (Figure 6C,C’). Interestingly, coinjection of human COL2A1 protein rescued the effects of the MO knockdown on Sox10 expression (Figure 6C’’’), suggesting evolutionarily and functionally conserved roles between human COL2A1 and zebrafish Col2a1a genes. Notably, the injection of human COL2A1 alone significantly decreased Sox10 expression in ocular and craniofacial neural crest cells (Figure 6D’). Taken together, these results suggest that Col2a1a is necessary for the maintenance of ocular and craniofacial Sox10 expression and the promotion of Sox10-positive neural crest cell migration during early eye development.

## 4. Discussion

The disorders observed in Stickler syndrome reflect mutations in the genes encoding instructions for the development and function of collagens as the major building blocks of connective tissue. The major clinical features of this disease include collagen-related bone disorders and craniofacial abnormalities, such as cleft palate, cleft lip, micrognathia and sensorineural or conductive hearing loss [1], as well as ocular manifestations, including vitreoretinal degeneration, myopia, cataracts, retinal detachments, crossed eyes, abnormal curvature to the cornea or lens and glaucoma [1,2,3,4,5,6]. In Stickler syndrome Type I, these defects are ascribed to autosomal dominant mutations in the COL2A1 gene [16,17,26] and can result in significant visual impairment or, in some cases, may lead to blindness. Type II collagen provides strength and structure to the connective tissues supporting the skin, organs, muscles, and joints of the body. COL2A1 is a well-known major component of the pro-alpha1(II) chain of type II collagen and an important building block of cartilage. Notably, COL2A1 is also expressed in neural crest-derived structures (i.e., vitreous humor, sclera, cornea, and trabecular meshwork) in the eye. As genetic causes of congenital anomalies often affect both craniofacial structures and anterior segment development [41,42,43,44], it would be prudent to characterize the potential role for COL2A1 in these tissues to address gaps in our knowledge of the role of this protein in eye development and gain further insight into the pathogenesis of Stickler syndrome Type I.

In the present study, we observed *col2a1a* expression in neural crest cells in the anterior segment structures from 48–60 hpf. This distinct cell population, along with the craniofacial neural crest, derives from the neural tube edge and subsequently migrates into the craniofacial region [36,37,45,46,47,48,49]. In previous studies, we showed that the neural crest cells destined for the anterior segment dorsoventrally migrate to the eye eventually settling in the periocular mesenchyme [28,33,50]. Although many Col2a1a-positive cells were detected in the dorsal-posterior quadrant between the surface ectoderm and optic cup, consistent with the craniofacial neural crest, *col2a1a* expression also marked a significant population of cells entering the anterior segment, showing marked expression in the iris outflow tract, sclera, and hyaloid. Col2a1a-positive cells were also detected in the ocular fissure, which provides a potential alternate pathway for entry into the anterior segment. Previous studies in mice, chickens, and humans do not describe an alternate neural crest migration pathway through the ocular fissure. However, ocular neural crest migration through the ocular fissure has recently been demonstrated in previous studies using time-lapse live imaging mediated through a transgenic zebrafish Foxd3 reporter line, which demarcates this distinct migratory cell population [28,51]. Classic studies on neural crest migration have described distinct waves of neural crest cells in humans and chickens (reviewed in [36,49] that contribute to the anterior segment (cornea, iris, and angle structures) in a spatiotemporal manner, but these waves have not yet been observed in zebrafish and mice. Notably, the presence of Col2a1a-positive cells in the anterior segment was only observed from 48–60 hpf and subsequently dissipated until *col2a1a* expression was only detected in the craniofacial population at 96 hpf. Additional studies are needed to determine the ultimate effect of this brief induction of *col2a1a* expression so early in development on the adult eye. Nevertheless, these studies were necessary for establishing Col2a1a as a marker for both ocular and craniofacial neural crest populations.

RA is a well-characterized regulator of embryonic craniofacial and ocular anterior segment formation [34,35,37,38,52,53]. Indeed, craniofacial and eye development requires highly regulated RA synthesis in the rhombencephalon and developing retina as well as the localized expression of the RA receptors (RARs) in the pharyngeal arches and periocular mesenchyme [22,54,55,56]. Moreover, our previous studies have confirmed RA as essential for early craniofacial and ocular neural crest migration and survival in zebrafish development from embryo to adulthood [42,44]. In the present study, we demonstrated *col2a1a* as a downstream target of RA regulation in the ocular neural crest. Reciprocal alterations in *col2a1a* expression were observed with both increased RA signaling through exogenous treatment with all-trans RA, which increased *col2a1a* in the ocular anterior segment and neural crest-derived jaw, and decreased RA activity through exogenous treatment with DEAB, which reduced *col2a1a* expression in these neural crest cell populations. These effects likely reflect the influence of RA on the ventral migration of the rhombencephalon neural crest, which derives the jaw and pharyngeal arches, and the dorsoventral migration of the mesencephalon/prosencephalon neural crest, which populates the periocular mesenchyme that subsequently gives rise to the ocular anterior segment [32,34,36,37,57]. Specifically, increased RA inhibits ventral neural crest migration, while decreased RA inhibits both dorsal and ventral neural crest migration [33]. With respect to *col2a1a*, which showed not only dorsoposterior expression but also dorsoventral expression, DEAB insult inhibits neural crest migration in both dorsal and ventral axes, limiting *col2a1a* expression to only the dorsal end of the axis and thereby decreasing overall *col2a1a* expression, particularly in the anterior segment of the eye. Conversely, with increased RA, *col2a1a* expression is limited ventrally but is unaffected dorsally and posteriorly, contributing to the observed increased expression of *col2a1a* in ocular, periocular and craniofacial regions. It is worthwhile to note that the potential alternative pathway for the entry of Col2a1a-expressing cells into the eye might also contribute to the observed increase in *col2a1a* expression in the presence of insults that inhibit the other routes. Further, since Pitx2 and RA are essential for the coordination of the signals that mediate ocular and craniofacial development [22,33,38], future studies should assess the influence of Pitx2 on *col2a1a* expression in the developing eye.

In the present study, like the effects of decreased RA signaling, 3% ETOH challenge decreased *col2a1a* expression in the anterior segment of the eye and developing jaw. Notably, in the setting of 3% ETOH, exogenous RA restored *col2a1a* expression in the anterior segment but not in the jaw. This finding is not surprising given that we have already demonstrated the regulation of *col2a1a* expression through RA in the ocular neural crest. In fact, RA has been implicated as a target for inhibition by ETOH through competition for binding to retinol dehydrogenase, the enzyme that mediates RA synthesis in this neural crest population [58]. Moreover, rescue of the effects of ETOH exposure on zebrafish through RA has previously been shown [59], although this study only focused on the neuroepithelial effect of ETOH exposure. Alternatively, in a previous study, we showed that ETOH exposure minimally affected the expression of transcription factors critical for neural crest migration and anterior segment formation, suggesting that ETOH may not significantly impact regulate retinoic acid signaling in the periocular mesenchyme [28], and the observed RA rescue of *col2a1a* expression may in part reflect the resistance of periocular and ocular neural crest cells to ETOH exposure compared to their craniofacial counterparts [28].

Indeed, studies in several models have previously shown that ethanol adversely affects the induction, expansion, migration, and differentiation of the cranial neural crest during development, and ETOH exposure increases the apoptosis of craniofacial neural crest cells [60,61,62,63,64]. Moreover, oxidative stress has previously been indicated as the mechanism underlying ETOH-induced apoptosis in the cranial neural crest [65,66,67,68]. Consistently, we also demonstrated that ETOH exposure increased oxidative stress in the cranial neural crest, however this effect was only observed in ventral neural crest cells that form the facial mesenchyme but not in those that contribute to the developing eye [28]. Notably, in this study, treatment with 3% ETOH starting at 24 hpf did significantly increase the percentage of apoptotic cells in the developing eye, but the cells undergoing apoptosis were primarily located in the retina and lens but not in the neural crest-derived ocular anterior segment [28]. Taken together, the results of these previous studies explain the apparent specificity of the RA-mediated rescue of *col2a1a* expression for the ocular neural crest and suggest that ethanol may directly affect *col2a1a* expression in ocular neural crest cells.

Consistent with the well-characterized contributions of Col2a1a to craniofacial development, knockdown of Col2a1a effectively decreased the formation of the neural crest-derived jaw and resulted in a smaller and bulged eye defect in zebrafish reminiscent of the clinical features of human patients with Stickler syndrome. Human COL2A1 showed improvement of the craniofacial and ocular defects due to MO-mediated Col2a1a knockdown, demonstrating the evolutionary conservation of COL2A1 function between humans and zebrafish and highlighting the importance of Col2a1a during early eye development. Importantly, in response to Col2a1a knockdown, Sox10-positive ocular anterior segment neural crest migration was disrupted, while Foxd3-positive anterior segment neural crest cell migration within the periocular mesenchyme and the optic fissure adjacent to the hyaloid vasculature remained consistent. Similarly, Col2a1a knockdown and overexpression adversely affected Sox10 expression in both the craniofacial and ocular neural crest, while Foxd3 expression remained unchanged, even with human COL2A1 overexpression. Preliminary fate mapping studies show minimal contributions of Sox10-positive cells to the eyes of adult zebrafish at 90 days postfertilization (dpf) (unpublished data), as these cells migrate from the anterior segment soon after optic cup formation is complete [51,69,70]. The loss of this early neural crest marker signals the expression of other transcription factors that, together with RA, promote ocular neural crest cell migration from the periocular mesenchyme and terminal differentiation [60]. The results of the present study suggest a mechanism whereby, the premature loss of Sox10 expression in response to Col2a1a knockdown or human COL2A1 overexpression traps Sox10-positive cells in the anterior segment and disrupts proper end cell differentiation into the anterior elements of the eye, such as the cornea, sclera, iris and trabecular meshwork [49,70]. Collagens are structural proteins located in the extracellular space of connective tissues that provide support to these cells. As components of the extracellular matrix, these proteins can either support or restrict neural crest cell migration [71], and their migration-promoting activity can be modulated by their conformation and/or association with other matrix components [49,71,72]. The results of the present study showing the adverse effects on Sox10 expression and Sox10-positive neural crest cell migration in response to Col2a1a knockdown or COL2A1 overexpression suggest a potential role for this protein in the promotion of Sox10-positive neural crest cell migration and terminal differentiation in the ocular neural crest.

Importantly, the knockdown of gene expression using MO translation blocking tools presents certain noteworthy limitations, including the transient effects of these oligos and associated knockdown effects, the timing of expression during development, and the specificity of the oligo for targeting the intended gene. Therefore, to fully appreciate the implications of the results obtained from the knockdown experiments in the present study, complementary studies implementing a stable model system using a highly precise gene editing tool are needed.

## 5. Conclusions

The present study assessed the role of Col2a1a in ocular neural crest cells that form the anterior segment of the eye. Consistent with Col2a1a localization in both the ocular and cranial neural crest cell populations and the functional effects of Col2a1a knockdown on the expression of Sox10 and migration of Sox10-positive craniofacial neural crest cells in the anterior segment, these studies suggest a permissive role for Col2a1a in the ocular neural crest to promote the precise migration of neural crest cells destined for the craniofacial region, thereby supporting the proper terminal differentiation of the anterior segment structures. These results provide further insight into the pathogenesis of Stickler syndrome, a congenital disease affecting both craniofacial and ocular development.

## Figures and Tables

**Figure 1 jdb-10-00042-f001:**
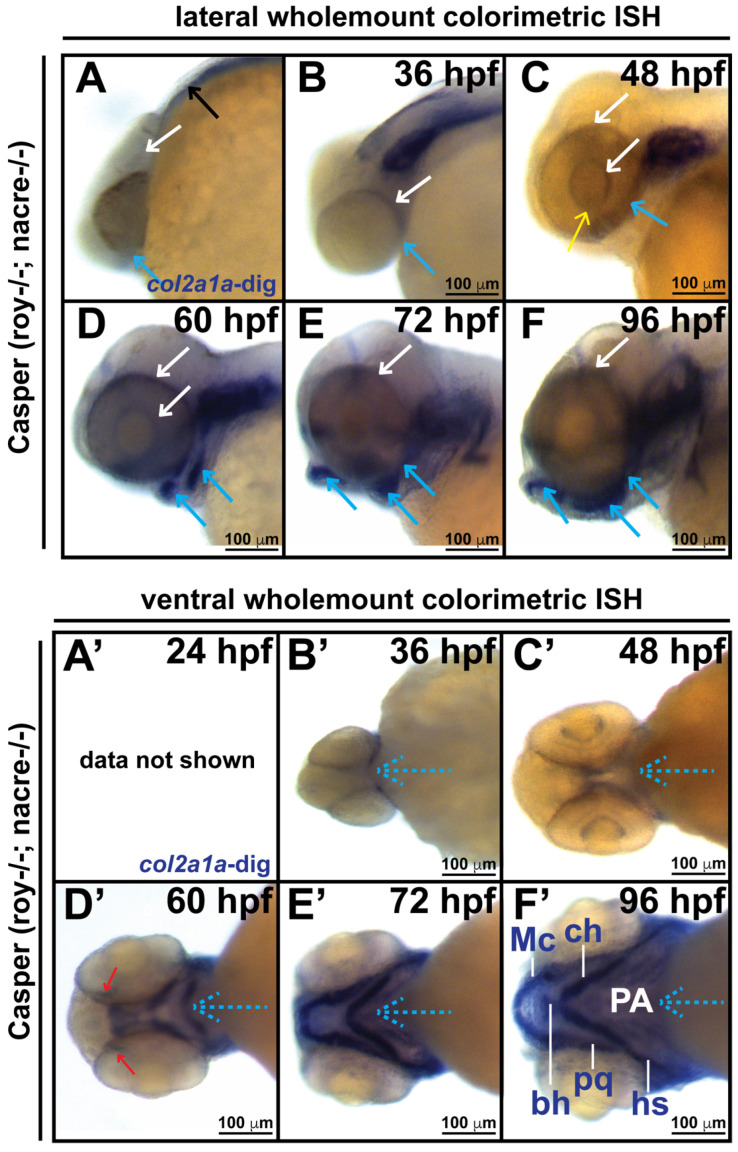
Col2a1a is expressed in the ocular neural crest during early development. Wholemount in situ hybridization in WT Casper (*roy−/−; nacre−/−*) zebrafish embryos during development at 24, 36, 48, 60, 72 and 96 hpf. Col2a1a gene expression was detected using a colorimetric assay (Vector Blue Substrate Kit, Vector Laboratories) that is both chromogenic (blue) and fluorescent (Far Red/Cy5). The sections were mounted in a media containing DAPI (gray). Lateral (**A**–**F**) brightfield wholemount images show that *col2a1a* expression initiates within the hindbrain (black arrow, (**A**) notochord at 24 hpf with dorsoposterior (white arrows, (**B**–**F**)) and ventral (blue arrows, (**B**–**F**)) progression into the ocular and craniofacial expression during eye and jaw development. The yellow arrow (**C**) highlights *col2a1a* expression in the ocular fissure at 48 hpf. Ventral (**B’**–**F’**) brightfield wholemount images show *col2a1a* expression in the developing jaw. The ventral-anterior progression of *col2a1a* expression is indicated (blue dashed arrows). The red arrows (**D’**) highlight *col2a1a* expression in the optic nerve at 60 hpf. By 96 hpf (**F’**), profound *col2a1a* expression was detected in the developing jaw and pharyngeal arches (PA). A ventral image of embryonic *col2a1a* expression at 24 hpf (**A’**) was not obtained because the embryo is tightly bound to the yolk mass at this stage of development, making it difficult to image the ventral side. Mc, Meckel’s cartilage; pq, palatoquadrate; ch, ceratohyal; bh, basihyal; hs, hyosympletic. Transverse cephalic sections and fluorescence (FL) microscopy analyses (**A”**–**F”**) provide additional information for significant ocular *col2a1a* expression in the anterior segment, with apparent expression in the anterior segment [iris (Ir), iris outflow tract (IOT), hyaloid vasculature (black arrows), sclera (Sc), periocular mesenchyme (PM), and optic nerve (ON)] at 48 hpf (**C”**). The dashed line in the lateral wholemount brightfield image of a 48 hpf embryo indicates the orientation of the plane of section, which passes perpendicular to the spinal column and extends in the rostral-caudal direction. (**G**,**H**) Wholemount colorimetric in situ hybridization, followed by GFP immunostaining in Tg(*sox10::EGFP*) and Tg(*foxd3::EGFP*) zebrafish embryos during early development at 48 hpf revealed the colocalization of *col2a1a* expression with neural crest cell markers in the developing jaw (**G**) [Meckel’s cartilage (Mc), trabeculae (T), and quadrate (Q)] and anterior segment (**H**) (hyaloid vasculature (white arrows), iris outflow tract (IOT), sclera (Sc), and periocular mesenchyme (PM)).

**Figure 2 jdb-10-00042-f002:**
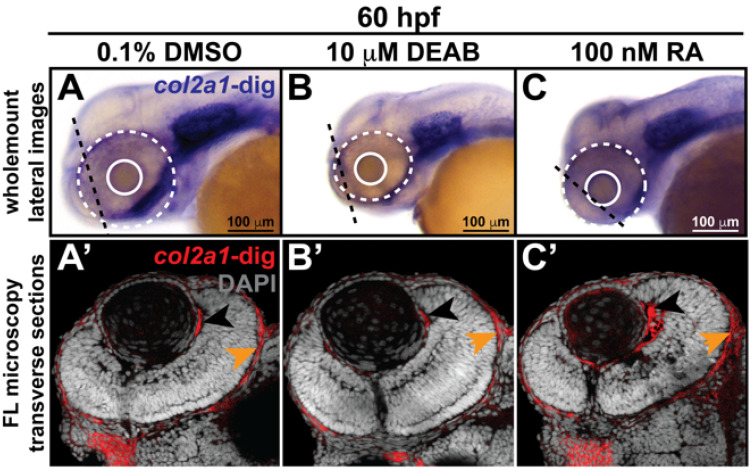
Retinoic acid regulates *col2a1a* expression in the ocular neural crest. Treatment with 10 mM DEAB, a selective pan-aldehyde dehydrogenase inhibitor, and 100 nM RA effectively decreased and increased, respectively, ocular *col2a1a* expression in WT Casper zebrafish embryos compared with their DMSO control-treated counterparts. Wholemount in situ analysis was performed using a chromogenic (blue) colorimetric assay (Vector Blue Substrate Kit, Vector Laboratories). Lateral images (**A**–**C**) at 60 hpf show the teratogenic effects of pharmacological insult on the ocular development of the treated embryos. The solid and dashed circles highlight the effect on eye size. The black dashed lines indicate the orientation of the plane of section, which passes perpendicular to the spinal column and extends in the rostral-caudal direction. FL microscopy analysis of transverse cephalic sections of the treated embryos at 60 hpf obtained after wholemount colorimetric in situ analysis developed using a fluorescent (Far Red/Cy5) substrate (Vector Blue Substrate Kit, Vector Laboratories) (**A’**–**C’**) show the effects of pharmacological insult on the anterior segment within the developing eye. The black arrowhead highlights *col2a1a* expression in the hyaloid vasculature and the orange arrowhead highlights *col2a1a* expression in the sclera/periocular mesenchyme. Quantification of these effects (**D**) shows that DEAB significantly decreased *col2a1a* expression, while RA significantly increased *col2a1a* expression. *, *p*-value ≤ 0.05. Ventral images (**E**,**F,G**) of wholemount in situ analysis at 90 hpf show the teratogenic effects of pharmacological insult on the ocular and craniofacial development of treated embryos. The solid and dashed circles highlight the effect on eye size. The white (Meckel’s cartilage development) and red (basihyal development) arrows highlight the effects on jaw development. The asterisks highlight the areas of pharyngeal arch development.

**Figure 3 jdb-10-00042-f003:**
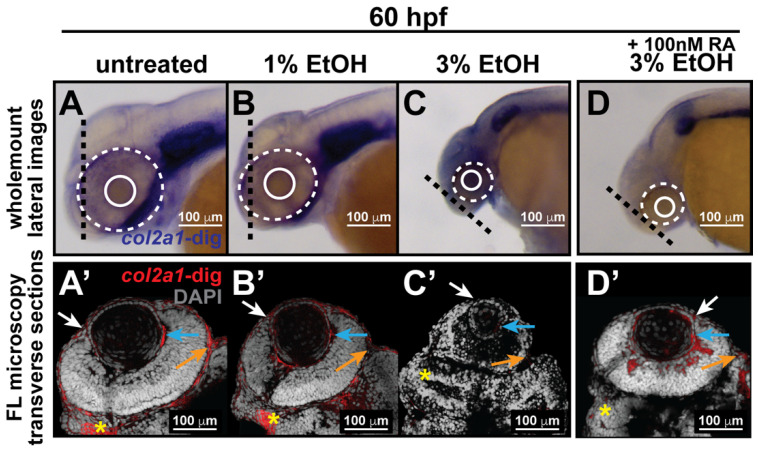
Ethanol decreased *col2a1a* expression in the ocular neural crest and developing jaw. Treatment with 1% and 3% ethanol from 24 to 48 hpf, followed by wholemount in situ hybridization analysis using a chromogenic (blue) colorimetric assay (Vector Blue Substrate Kit, Vector Laboratories) and FL microscopy analysis of the fluorescent (Far Red/Cy5) colorimetric substrate (Vector Blue Substrate Kit, Vector Laboratories) at 60 and 90 hpf revealed decreased *col2a1a* expression in the developing eyes and jaws of treated WT Casper zebrafish embryos compared with their untreated and 1% ETOH-treated counterparts. Lateral images (**A**–**C**) of wholemount colorimetric in situ analysis at 60 hpf show the teratogenic effects on the ocular development of the treated embryos. The solid and dashed circles highlight the effect on eye size. The black dashed lines indicate the orientation of the plane of section, which passes perpendicular to the spinal column and extends in the rostral-caudal direction. Transverse cephalic sections of the colorimetric in situ analysis (**A’**–**C’**) of treated embryos at 60 hpf show a significant decrease in *col2a1a* expression in the anterior segment [white arrow, iris; blue arrow, hyaloid; orange arrow, sclera, and periocular mesenchyme) of the developing eye with 3% ETOH treatment. The yellow asterisks highlight craniofacial *col2a1a* expression. Quantification of these effects (**E**) shows that 3% significantly decreased *col2a1a* expression. *, *p*-value ≤ 0.05. Ventral images (**F–I**) of wholemount colorimetric in situ analysis show significantly delayed jaw development, with severe malformations of the pharyngeal arches (PA) and basihyal (red arrow) and Meckel’s cartilage (white arrow) in treated embryos at 96 hpf. Combined treatment with 3% ETOH from 24–48 hpf and RA from 27–96 hpf increased *col2a1a* expression in the ocular anterior segment (**D’**) and partially rescued the teratogenic effects of 3% ethanol treatment on eye development at 60 hpf (**D**,**E**) but did not restore *col2a1a* expression in the craniofacial neural crest (yellow asterisk, (**D’**)) and further potentiated the deleterious effects of ethanol treatment on the developing jaw (white asterisk, (**I**)) at 96 hpf.

**Figure 4 jdb-10-00042-f004:**
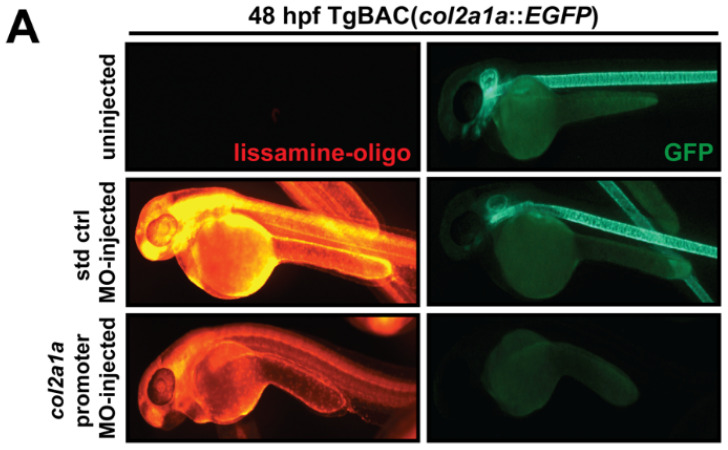
Col2a1a knockdown mimicked the characteristic features of human Stickler syndrome in zebrafish. A lissamine-tagged antisense MO targeting the promoter region (prMO) of *col2a1a* was injected into TgBAC(*col2a1a::EGFP*) and Tg(*sox10::EGFP*) embryos at the single-cell stage. Lateral FL microscopy images (**A**) show the loss of GFP reporter expression driven by the *col2a1a* promoter in TgBAC(*col2a1a::EGFP*) transgenic fish (lower right panel) compared with uninjected and standard control (std ctrl) MO-injected embryos (upper right and middle panels, respectively). Quantification of this effect showed a significant loss of expression (**B**). MO-injected Sox10-positive transgenic zebrafish embryos showed skeletal defects, including scoliosis (upper right panel, dashed black line, (**C**)) and significantly shortened body length (**D**), measured perpendicular to the spine from the tip of the epiphysis to the area above the urogenital opening (white dashed arrow (inset, **D**), delayed neural crest-derived jaw development (white arrowhead, upper right panel and black asterisk, lower right-most panel, (**C**)), and significantly smaller bulging eyes (solid and dashed circles, (**C**,**E**)), as measured along the anterior–posterior (a-p; inset, **E**) and dorsal-ventral (d-v; inset, **E**) axes, compared with uninjected and std ctrl MO-injected embryos at 48 and 72 hpf (left and middle panels, (**C**–**E**)). *, *p*-value ≤ 0.05; **, *p*-value ≤ 0.001.

**Figure 5 jdb-10-00042-f005:**
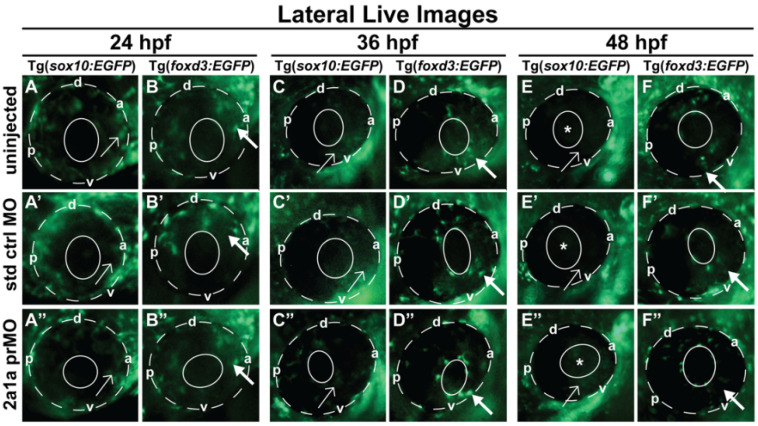
Col2a1a MO knockdown differentially affected Sox10 and Foxd3 expression and ocular neural crest cell migration during early eye development. Lateral live imaging at 24, 36 and 48 hpf showing the effects of Col2a1a knockdown on GFP reporter expression in Tg (*foxd3::EGFP*) and Tg(*sox10::EGFP*) transgenic zebrafish embryos injected at the single-cell stage with a lissamine-tagged antisense prMO targeting *col2a1a*. (**A**–**F**) uninjected, (**A’**–**F’**) standard control (std ctrl) MO-injected, (**A”**–**F”**) *col2a1a* promoter MO (prMO)-injected. Foxd3-positive cell migration (bold arrows, (**F”**)) was not affected after MO injection compared with that in std ctrl MO-injected (**D”**) and uninjected embryos (**B”**). Sox10-positive cell migration (**E”**) was markedly disrupted in response to Col2a1a MO knockdown, as these cells were retained in the anterior segment (asterisk) and periocular mesenchyme (single arrows) in contrast to the cells in their uninjected (**F**) and std ctrl-injected counterparts (**F’**), which showed few Sox10-positive cells in the anterior segment at 48 hpf. The solid and dashed circles highlight the lens and retinal piment epithelium, respectively, in the zebrafish eyes. d, dorsal; v, ventral; p, posterior; a, anterior.

**Figure 6 jdb-10-00042-f006:**
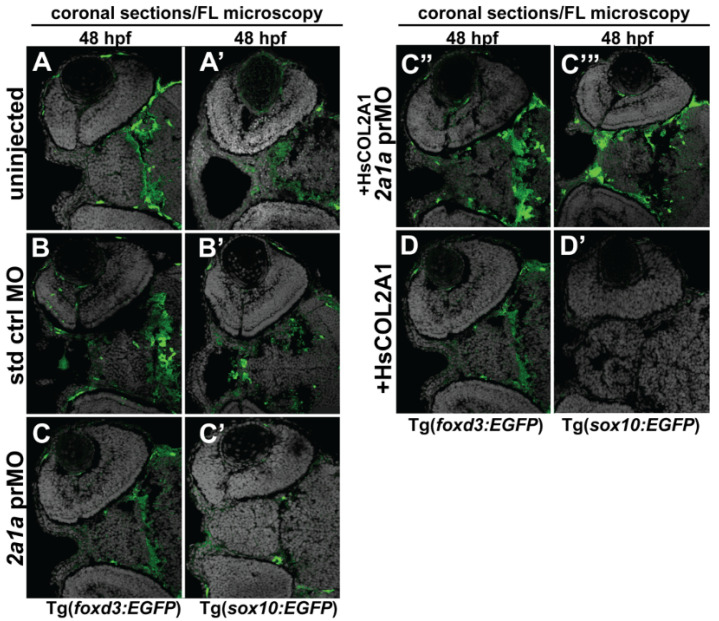
Col2a1a MO knockdown differentially affected ocular neural crest cell Sox10 and Foxd3 expression during early eye development. FL microscopy analysis of the effects of Col2a1a knockdown on GFP reporter expression in transverse cephalic sections from 48 hpf Tg (*foxd3::EGFP*) and Tg(*sox10::EGFP*) transgenic zebrafish embryos injected at the single-cell stage with a lissamine-tagged antisense prMO targeting *col2a1a*. MO knockdown of Col2a1a adversely affected Sox10-positive GFP expression in ocular anterior segment neural crest cells (**A’**–**C’**). The Foxd3-positive GFP expression in ocular neural crest cells was not affected by Col2a1a knockdown (**A**–**C**,**C”**,**D**). The coinjection of human (Hs) COL2A1 protein rescued Sox10-positive GFP expression (**C’’’**), while the injection of HsCOL2A1 alone (**D’**) adversely affected ocular Sox10-positive GFP expression in neural crest cells.

## Data Availability

Not applicable.

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
