# Peer review of "Zebrafish Model of Stickler Syndrome Suggests a Role for Col2a1a in the Neural Crest during Early Eye Development"

_jdb, 2022, doi:10.3390/jdb10040042_

Round 1

Reviewer 1 Report

Williams and Bohnsack have investigated thxe role of col2a1a in neural crest migration and differentiation during early Zebrafish eye development. The authors first examine col2a1a expression around the developing eye between 24 hpf and 96 hpf by in situ hybridization. They then treat embryos with DEAB to inhibit retinoic acid production, or with exogeneous retinoic acid and describe that col2a1a is a downstream target of retinoic acid in cranial and ocular neural crest cells. Next, they treated embryos with 1% and 3% ethanol and describe a strong downregulation of col2a1a and severe morphological defects after 3% ethanol treatment. They then examined the effect of a combined ethanol and RA treatment and find a partial rescue of the morphological eye defects but not of col2a1 expression and jaw defects. They show that col2a1a colocalizes with the neural crest cell markers Sox10 and Foxd3 in the developing eye. To examine the function of col2a1a the authors used a translation blocking morpholino to knockdown col2a1a and describe craniofacial and ocular defects that resemble defects in human Stickler syndrome. Finally, the authors describe an altered distribution of Sox10 positive cells in the eye of col2a1a knock down embryos.

The data that the author present in their manuscript are interesting and relevant to the understanding of human Stickler syndrome. However, there are a number of points that the authors need to address prior to publication:

1.     In section 3.2. the authors should add (a) reference(s) for their statement made in sentence 1 and also for the DEAB inhibitor (the latter references(s) could also be incorporated into “Material and Methods”).

2.     It remains unclear whether ethanol treatment affects col2a1a expression in neural crest cells in the cranial area or whether the decreased col2a1a in situ signal is the consequence of a reduced number of neural crest cells in that area due to changes in neural crest migration or survival. The authors should include pictures of treated embryos stained with neural crest cell markers in their figures 2, 3 and 4.

3.     The authors have chosen “Col2a1a regulates expression and migration of Sox10 positive craniofacial neural crest cells in the ocular segment…. “ as the heading for section 3.6. However, the data they present does not allow to distinguish whether the changes in the Sox10 staining pattern are the consequence of alterations in Sox10 expression in neural crest cells in the eye or a changes in the migration of those Sox10 positive cells. In addition the grammar of the heading is wrong (“Col2a1a regulates expression … of Sox10 positive craniofacial neural crest cells…” - that doesn’t make sense!).

4.     In figures 1-4, the authors show sections of in situ hybridizations with a red “Cy5” staining, that appear to be fluorescent images. The authors should explain in “Material and Methods” or in the figure legends how these images were generated and what they show here. Why is the chromogenic blue in situ signal shown in red, and is the grey tissue color due to auto-fluorescence?

5.     In the legend to figure 2 an explanation of the abbreviations used to label anatomical structures in the pictures is missing. In addition, the labels are too small, pixelated and due to the chosen color hard to see.

6.     In figures 2A-C and 3 A-C, the embryo heads differ a lot in size. Have these pictures been taken at the same magnification? The figures should contain scale bars to avoid this unclarity.

7.     The pictures shown in figure 4 should be incorporated into figure 3 so that it is possible to directly compare the effects of ethanol treatment alone with the effect of a combined ethanol and RA treatment.

8.     Spelling: line 35: “virtual degeneration” should be “vitreal degeneration”

Author Response

The authors would like to thank you for your kind support and rigorous examination of this work. We appreciate your comments and wholeheartedly agree with your suggestions for the revision of the manuscript. Based on your comments and concerns we have made extensive edits to much of the manuscript and believe that the revised version is now more focused and persuasive.

Please find below the point-by-point responses to your comments.

Reviewer #1

  1. In section 3.2. the authors should add (a) reference(s) for their statement made in sentence 1 and also for the DEAB inhibitor (the latter references(s) could also be incorporated into “Material and Methods”).

Appropriate references supporting the statements mentioned here have been added to the text.

  1. It remains unclear whether ethanol treatment affects col2a1a expression in neural crest cells in the cranial area or whether the decreased col2a1a in situ signal is the consequence of a reduced number of neural crest cells in that area due to changes in neural crest migration or survival. The authors should include pictures of treated embryos stained with neural crest cell markers in their figures 2, 3 and 4.

Several studies have previously demonstrated the multiple adverse effects on the cranial neural crest in response to ethanol exposure. These effects range from defects in the induction, expansion, apoptosis, migration, and differentiation of cranial neural crest cells to increased oxidative stress as a mechanism of apoptosis in these cells. Although we agree that these factors may indeed account for the loss of col2a1a expression in the craniofacial region, further exploration is beyond the scope of this paper, which focuses on col2a1a contributions to the ocular neural crest (anterior segment). To this end, we did not repeat any of the experiments to include pictures of treated embryos stained with neural crest cell markers, but we did provide a brief discussion of these multiple adverse effects, which likely reflect the lack of col2a1a expression in the craniofacial neural crest.

Genetic factors further modify vulnerability to ethanol-induced craniofacial dysmorphology and include genes important for neural crest development, including shh signaling, PDFGA, vangl2, and ribosomal biogenesis.

  1. The authors have chosen “Col2a1a regulates expression and migration of Sox10 positive craniofacial neural crest cells in the ocular segment…. “ as the heading for section 3.6. However, the data they present does not allow to distinguish whether the changes in the Sox10 staining pattern are the consequence of alterations in Sox10 expression in neural crest cells in the eye or a changes in the migration of those Sox10 positive cells. In addition the grammar of the heading is wrong (“Col2a1a regulates expression … of Sox10 positive craniofacial neural crest cells…” - that doesn’t make sense!).

This heading has been changed to “Col2a1a MO knockdown differentially affected Sox10- and Foxd3-positive neural crest cells during early eye development”, which now more accurately reflects the data presented in the current paper.

  1. In figures 1-4, the authors show sections of in situ hybridizations with a red “Cy5” staining, that appear to be fluorescent images. The authors should explain in “Material and Methods” or in the figure legends how these images were generated and what they show here. Why is the chromogenic blue in situ signal shown in red, and is the grey tissue color due to auto-fluorescence?

A description of the probe detection and subsequent colorimetric analysis has been added to the Materials and Methods. Briefly, the probe was detected using an alkaline phosphatase-labeled anti-digoxigenin antibody. The AP activity was detected using a colorimetric assay that is both chromogenic (blue) and fluorescent (Far Red/Cy5). After subsequent cryoprotection and sectioning, the sections were mounted in a media containing DAPI, which was pseudo colored from blue to gray/white in the figures to make the fluorescence in the Far-red channel even more visible when overlaid onto the DAPI background.   

  1. In the legend to figure 2 an explanation of the abbreviations used to label anatomical structures in the pictures is missing. In addition, the labels are too small, pixelated and due to the chosen color hard to see.

Presentation of the data in Figure 2 has been revised and the figure legend, which now includes an explanation of the abbreviations used to label the anatomical structures in the pictures, has been rewritten.

  1. In figures 2A-C and 3 A-C, the embryo heads differ a lot in size. Have these pictures been taken at the same magnification? The figures should contain scale bars to avoid this unclarity.

The pictures of the embryo heads shown in Figures 2 and 3 were captured at the same magnification and edited using the same parameters. The differences in size reflect delayed development in response to RA/DEAB and ethanol challenge. For clarification, scale bars have been added to all figures where appropriate.

  1. The pictures shown in figure 4 should be incorporated into figure 3 so that it is possible to directly compare the effects of ethanol treatment alone with the effect of a combined ethanol and RA treatment.

The data shown in Figures 3 and 4 have been combined and the figures have been revised to allow for the direct comparison of the effects of combined ethanol and RA treatments.

  1. Spelling: line 35: “virtual degeneration” should be “vitreal degeneration”

This spelling correction has been applied

Reviewer 2 Report

In this paper entitled “Zebrafish model of Stickler syndrome suggests a role for Col2a1a in the neural crest during early eye development” the authors showed that Col2a1a is expressed in the neural crest-derived elements of the developing zebrafish eye and that Col2a1a MO-KD causes defects that resemble Stickler syndrome. They also show that RA treatment increases Co2a1a while EtOH treatment decrease it expression in the developing eye. While this work is relevant and could have a high impact in the field, the lack of quantification of findings and statistical analysis decreases the relevancy of findings as changes can be highly variable in an animal model such as zebrafish.

Major points are:

1)    Quantification of results (coronal sectional ISH/colorimetric microscopy; migration assay): the authors write several times that they detect significant changes without quantification (lines 169, 192, 249, 290, 292). This is not acceptable when using a model system as zebrafish in which changes can be variable between embryos. Experiments were repeated 4 to 6 times using 50 to 100 embryos per group as described in the method section. Thus, it is possible to perform quantification followed by statistical analysis that will determine if a result is significant or not.

2)    Lines 200-202 (Altogether, these results confirm col2a1a expression in periocular and ocular neural crest populations in the developing eye, which are less sensitive to ethanol than the craniofacial neural crest cell population that derives the jaw and pharyngeal arches) does not seem to summarize the results just described. The authors showed that ethanol treatment disrupted eye and jaw development with defects on both regions being more severe at 3%. Then they show that RA treatment primarily rescued the eye defects while making jaw defects more severe. This sentence needs to be rewritten to match the findings.

3)    Lines 196, 299: the authors state that there is an “increase” in col2a1 expression in the rescue experiment (3% ethanol + RA). However, because figure 4 lacks a control untreated and 3% ethanol treated zebrafish embryos, it is hard to make a call if this is an increased expression (higher than control untreated expression) or just a rescued expression (higher than 3% ethanol but lower or similar to control untreated levels). Based on figure 3A’, expression of Col2a1 in Figure 4A’ is not higher (increased) than 3A’ in all areas of the developing eye. Finally, quantification of the expression changes is lacking (major point 1) and thus why the results are not convincing as it is (qualitative analysis).

4)    Figures 1, 2, 3,4: the legend lacks explanation for Figure 1A’’-F”. What is Cy5 showing? It is usually common to write on the figure what the fluorochrome is marking (col2a1) rather than the name of the fluorochrome itself. This information would be located in the method section. However, in the method section there is not mention of Cy5 being used; rather they described immunostaining with Alexa Fluor 488 (line 126). The figure is also not mentioning they are likely using the TgBAC (cola2a1::EGFP). I am assuming they are using it since there is not mention whatsoever in this legend or the figure legend of any subsequent figure that likely uses this line.

5)    Figures 1, 2, 3, 4: because the authors add whole-mount images, it would be very informative if they showed a plane of section (dashed lines) for the coronal section on the whole-mounts for orientation.

6)    Figures 1, 2, 3, 4: the authors should add arrows/arrowhead to point the reader where to look, particularly in figures 2, 3 and 4, where they are describing changes in expression after a treatment.

7)    Figure 6: Another figure lacking some sort of quantification. Usually research will use a scoring system to quantify defects especially because they can be highly variable.

8)    Figure 8 legend – lines 335-339 (MO knockdown of Col2a1 decreased the migration of sox10-positive ocular anterior segment neural crest cells. (A-C, C”, D) The migration of foxd3-positive ocular neural crest cells was not affected by Col2a1a knockdown. (C”’) Coinjection of human COL2A1 protein rescued the effects of the MO knockdown on sox10-positive cell migration). The experiment is not showing changes in migration but changes in expression of markers (foxd3; sox10) that can be caused by changes in migration. Another possible explanation could be changes in cell death or proliferation. Thus, the authors need to rewrite this legend and corresponding section 3.6.

9)    Cell death and proliferation: the authors do not mention anywhere in the discussion that changes could be cause by alterations in cell death or proliferation. Either add the assay with quantification or add a paragraph in the discussion about this possibility.

10) Discussion lines 441-444 (Thus, a key function of Col2a1a in the ocular neural crest may be to maintain the expression of Sox10 in the anterior segment to ensure the complete migration of Sox10-positive craniofacial neural crest cells out of this ocular region prior to the establishment of the anterior segment structures). Col2a1a is not a transcription factor; thus it cannot be directly maintaining expression of Sox10. It is not regulating expression inside the cell either since it is a component of the extracellular matrix. There is not discussion about these points. The authors need to add a paragraph with potential ways col2a1 is regulating expression of Col2a1a (integrins???).

Minor points:

1.    Figure 5 (section 3.4) seems misplaced since it is another descriptive figure similar to figure 1. As a suggestion, I would either merge figure 5 with figure 1 or move it to figure 2.

2.    Discussion – lines 394-398: this is a rehash of the result section. The authors should add what the significance of these findings is rather than just repeat the findings.

Author Response

The authors would like to thank you for your kind support and rigorous examination of this work. We appreciate your comments and wholeheartedly agree with your suggestions for the revision of the manuscript. Based on your comments and concerns we have made extensive edits to much of the manuscript and believe that the revised version is now more focused and persuasive.

Please find below the point-by-point responses to your comments.

Reviewer #2

Major points:

1)Quantification of results (coronal sectional ISH/colorimetric microscopy; migration assay): the authors write several times that they detect significant changes without quantification (lines 169, 192, 249, 290, 292). This is not acceptable when using a model system as zebrafish in which changes can be variable between embryos. Experiments were repeated 4 to 6 times using 50 to 100 embryos per group as described in the method section. Thus, it is possible to perform quantification followed by statistical analysis that will determine if a result is significant or not.

Quantification of the results has been conducted and these data have been incorporated into the manuscript where appropriate (Methods, Figures and Results).

2)Lines 200-202 (Altogether, these results confirm col2a1a expression in periocular and ocular neural crest populations in the developing eye, which are less sensitive to ethanol than the craniofacial neural crest cell population that derives the jaw and pharyngeal arches) does not seem to summarize the results just described. The authors showed that ethanol treatment disrupted eye and jaw development with defects on both regions being more severe at 3%. Then they show that RA treatment primarily rescued the eye defects while making jaw defects more severe. This sentence needs to be rewritten to match the findings.

The results summary for this section has been rewritten as follows: “Altogether, these results confirm show that ethanol treatment inhibited col2a1a expression in periocular and both ocular and craniofacial neural crest populations in the developing embryo and disrupted eye and jaw development, with defects in both regions being more severe with 3% ETOH challenge. Notably, RA significantly restored ocular col2a1a expression and primarily rescued the eye defects but made the jaw defects more profound”

3)Lines 196, 299: the authors state that there is an “increase” in col2a1 expression in the rescue experiment (3% ethanol + RA). However, because figure 4 lacks a control untreated and 3% ethanol treated zebrafish embryos, it is hard to make a call if this is an increased expression (higher than control untreated expression) or just a rescued expression (higher than 3% ethanol but lower or similar to control untreated levels). Based on figure 3A’, expression of Col2a1 in Figure 4A’ is not higher (increased) than 3A’ in all areas of the developing eye. Finally, quantification of the expression changes is lacking (major point 1) and thus why the results are not convincing as it is (qualitative analysis).

The data shown in Figures 3 and 4 have been combined and the figures have been revised to allow for the direct comparison of the effects of combined ethanol and RA treatments. Quantification of the data been conducted.

4)Figures 1, 2, 3,4: the legend lacks explanation for Figure 1A’’-F”. What is Cy5 showing? It is usually common to write on the figure what the fluorochrome is marking (col2a1) rather than the name of the fluorochrome itself. This information would be located in the method section. However, in the method section there is not mention of Cy5 being used; rather they described immunostaining with Alexa Fluor 488 (line 126). The figure is also not mentioning they are likely using the TgBAC (cola2a1::EGFP). I am assuming they are using it since there is not mention whatsoever in this legend or the figure legend of any subsequent figure that likely uses this line.

An explanation for Figure 1A’’-F” has been added, and the text in the entire legend has been revised.

A description of the probe detection and subsequent colorimetric analysis has been added to the Materials and Methods. Briefly, the probe was detected using an alkaline phosphatase-labeled anti-digoxigenin antibody. The AP activity was detected using a colorimetric assay that is both chromogenic (blue) and fluorescent (Far Red/Cy5). After subsequent cryoprotection and sectioning, the sections were mounted in a media containing DAPI, which was pseudo colored from blue to gray/white in the figures to make the fluorescence in the Far-red channel even more visible when overlaid onto the DAPI background.  

The WT Casper strain was used for the in situ hybridization experiment shown in Fig 1 and the probe was detected as described above. For clarity, descriptions of the lines used for the in situ hybridization and MO knockdown experiments have been added to the Methods, figures, legends and main body where appropriate.

5)Figures 1, 2, 3, 4: because the authors add whole-mount images, it would be very informative if they showed a plane of section (dashed lines) for the coronal section on the whole-mounts for orientation.

The plane of section (dashed lines) for the transverse cephalic sections has been added on the wholemounts for orientation.

6)Figures 1, 2, 3, 4: the authors should add arrows/arrowhead to point the reader where to look, particularly in figures 2, 3 and 4, where they are describing changes in expression after a treatment.

Arrows, arrowheads, asterisks, and other indicators have been added to the figures to direct the reader where to look when changes in the expression after a treatment have been described.

7)Figure 6: Another figure lacking some sort of quantification. Usually research will use a scoring system to quantify defects especially because they can be highly variable.

Notably, there is no “scoring system” used in zebrafish research, as the developmental defects are not assessed through behaviors and postures, as in mice, but rather can easily visualized in these transparent animal models, facilitating real-time morphological observations in the animals through microscopic analyses. To this end, I have revised the data for Figure 5 (formerly Figure 6) has been quantified and the figure presentation and discussion of the results has been revised.

8)Figure 8 legend – lines 335-339 (MO knockdown of Col2a1 decreased the migration of sox10-positive ocular anterior segment neural crest cells. (A-C, C”, D) The migration of foxd3-positive ocular neural crest cells was not affected by Col2a1a knockdown. (C”’) Coinjection of human COL2A1 protein rescued the effects of the MO knockdown on sox10-positive cell migration). The experiment is not showing changes in migration but changes in expression of markers (foxd3; sox10) that can be caused by changes in migration. Another possible explanation could be changes in cell death or proliferation. Thus, the authors need to rewrite this legend and corresponding section.

The legend for Figure 8 (Fig. 6 in revised manuscript) and the corresponding section (3.5) have been rewritten.

9)Cell death and proliferation: the authors do not mention anywhere in the discussion that changes could be cause by alterations in cell death or proliferation. Either add the assay with quantification or add a paragraph in the discussion about this possibility.

A discussion of the possibility of cell death and proliferation in response to ETOH challenge has been added.

10)Discussion lines 441-444 (Thus, a key function of Col2a1a in the ocular neural crest may be to maintain the expression of Sox10 in the anterior segment to ensure the complete migration of Sox10-positive craniofacial neural crest cells out of this ocular region prior to the establishment of the anterior segment structures). Col2a1a is not a transcription factor; thus, it cannot be directly maintaining expression of Sox10. It is not regulating expression inside the cell either since it is a component of the extracellular matrix. There is not discussion about these points. The authors need to add a paragraph with potential ways col2a1 is regulating expression of Col2a1a (integrins???).

A discussion of the potential role for Col2a1a in the ocular neural crest, appropriately accounting for its

Minor points:

  1. Figure 5 (section 3.4) seems misplaced since it is another descriptive figure similar to figure 1. As a suggestion, I would either merge figure 5 with figure 1 or move it to figure 2.

Figure 5 has been revised and merged with Figure 1. In addition, the text corresponding to this data has been revised and moved to the appropriate location in the document consistent with the discussion of the other descriptive data in Figure 1.

  1. Discussion – lines 394-398: this is a rehash of the result section. The authors should add what the significance of these findings is rather than just repeat the findings.

A discussion of the significance of these findings has now been added to the text in this section.

Round 2

Reviewer 1 Report

The authors have successfully made an effort to address the reviewers' comments and the manuscript is now much better. I therefore have no hesitations to recommend publication. Nevertheless there are a view minor text errors that should be corrected prior to publication: 

1. In the first paragraph of the results  it says:  "... and progresses dorsoposteriorally and ventrally into the ocular (white arrows, Fig. 1A-F) .." Problem: There is no white arrow in Fig. 1F.

2. In the first paragraph of the results it also say:  "with apparent ex- pression in the anterior segment [iris (Ir) (Fig. 1C”), hyaloid vasculature (black arrows, Fig. 1C”), sclera (Sc) and periocular mesenchyme (PM) (Fig. 1C”), ocular fissure (white yellow arrowhead, Fig. 1D’1C), and optic nerve (ON) (red arrows, Fig. 1D’; Fig. 1C”)] at 48 and 60 hpf (Fig. 1D”)" 

Problem: There is no Fig. 1 C'' , D'' or F''. And there is no white yellow arrowhead.

Author Response

  1. In the first paragraph of the results  it says:  "... and progresses dorsoposteriorally and ventrally into the ocular (white arrows, Fig. 1A-F) .." Problem: There is no white arrow in Fig. 1F.

Thank you for catching this oversight. A white arrow has been added to Fig. 1F

  1. In the first paragraph of the results it also say:  "with apparent expression in the anterior segment [iris (Ir) (Fig. 1C”), hyaloid vasculature (black arrows, Fig. 1C”), sclera (Sc) and periocular mesenchyme (PM) (Fig. 1C”), ocular fissure (white yellow arrowhead, Fig. 1D’1C), and optic nerve (ON) (red arrows, Fig. 1D’; Fig. 1C”)] at 48 and 60 hpf (Fig. 1D”)" 

Problem: There is no Fig. 1 C'' , D'' or F''. And there is no white yellow arrowhead.

"with apparent expression in the anterior segment [iris (Ir) (Fig. 1C”), hyaloid vasculature (black arrows, Fig. 1C”), sclera (Sc) and periocular mesenchyme (PM) (Fig. 1C”), ocular fissure (white yellow arrowhead, Fig. 1D’1C), and optic nerve (ON) (red arrows, Fig. 1D’; Fig. 1C”)] at 48 and 60 hpf (Fig. 1D”)"

Thank you for catching this oversight. The revised version of this figure was accidentally left out of the revised manuscript. Figure 1 has now been completely revised and all information has been added. The text in the main document has been revised as follows: "with apparent expression in the anterior segment [iris (Ir) (Fig. 1C”), hyaloid vasculature (black arrows, Fig. 1C”), sclera (Sc) and periocular mesenchyme (PM) (Fig. 1C”), ocular fissure (yellow arrow, Fig. 1C), and optic nerve (ON) (red arrows, Fig. 1D’; Fig. 1C”)] at 48 and 60 hpf (Fig. 1D”)"

Reviewer 2 Report

The authors addressed all major points particularly quantification of results. This article is now suitable for publication.

Author Response

Thank you for your support. Your Round 1 comments helped to markedly improve the suitability of this manuscript.